# WHICH SHORTCUT CUES WILL DNNS CHOOSE? A STUDY FROM THE PARAMETER-SPACE PERSPECTIVE

**Luca Scimeca**\*, **Seong Joon Oh**\*, **Sanghyuk Chun, Michael Poli & Sangdoo Yun**
NAVER AI Lab, NAVER Corp
Seoul, 13638, Republic of Korea
`luca.scimeca@dfci.harvard.edu`

## ABSTRACT

Deep neural networks (DNNs) often rely on easy–to–learn discriminatory features, or *cues*, that are not necessarily essential to the problem at hand. For example, ducks in an image may be recognized based on their typical background scenery, such as lakes or streams. This phenomenon, also known as *shortcut learning*, is emerging as a key limitation of the current generation of machine learning models. In this work, we introduce a set of experiments to deepen our understanding of shortcut learning and its implications. We design a training setup with *several* shortcut cues, named `WCST-ML`, where each cue is equally conducive to the visual recognition problem at hand. Even under equal opportunities, we observe that (1) certain cues are preferred to others, (2) solutions biased to the easy–to–learn cues tend to converge to relatively flat minima on the loss surface, and (3) the solutions focusing on those preferred cues are far more abundant in the parameter space. We explain the abundance of certain cues via their Kolmogorov (descriptional) complexity: solutions corresponding to Kolmogorov-simple cues are abundant in the parameter space and are thus preferred by DNNs. Our studies are based on the synthetic dataset DSprites and the face dataset UTKFace. In our `WCST-ML`, we observe that the bias of models leans toward simple cues, such as color and ethnicity. Our findings emphasize the importance of active human intervention to remove the model biases that may cause negative societal impacts.

## 1 INTRODUCTION

Emerging studies on the inner mechanisms of deep neural networks (DNNs) have revealed that many models have *shortcut biases* (Cadene et al., 2019; Weinzaepfel & Rogez, 2021; Bahng et al., 2020; Geirhos et al., 2020). DNNs often pick up simple, non-essential cues, which are nonetheless effective within a particular dataset. For example, a DNN trained for the task of animal recognition may recognize ducks while attending on water backgrounds, given the strong correlations between such background cues and the target label (Choe et al., 2020). These shortcut biases often result in a striking qualitative difference between human and machine recognition systems; for example, convolutional neural networks (CNNs) trained on ImageNet extensively rely on texture features, while humans would preferentially look at the global shape of objects (Geirhos et al., 2019). In other cases, the shortcut bias arises in models that suppress certain streams of inputs: visual question answering (VQA) models often neglect the entire image cues, for one does not require images to answer questions like "what color is the banana in the image?" (Cadene et al., 2019)

Since DNNs have successfully outperformed humans on many tasks (Silver et al., 2016; Rajpurkar et al., 2017), such a phenomenon may look benign at face value. However, the shortcut biases become problematic when it comes to the generalization to more challenging test-time conditions, where the shortcuts are no longer valid (Cadene et al., 2019; Weinzaepfel & Rogez, 2021; Bahng et al., 2020; Geirhos et al., 2020). These biases also cause ethical concerns when the shortcut features adopted by a model are sensitive like gender or skin color (Wang et al., 2019; Xu et al., 2020).

Instead of proposing a method or solution, this work focuses on deepening our understanding of the shortcut bias phenomena. In particular, we design a dataset where multiple cues are *equally valid*

---

\*First two authors contributed equally.

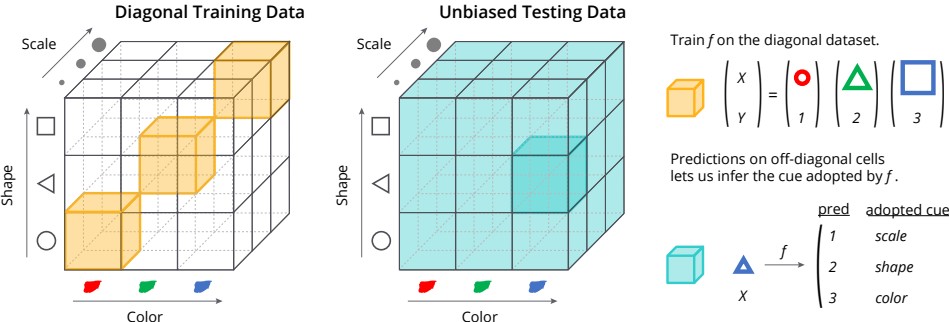

Figure 1: **Wisconsin Card Sorting Test for Machine Learners (`WCST-ML`).** The training dataset (left) poses *equally plausible* cues to the DNN: color, shape, and scale. Which cue will a model $f$ choose to use, when given such training data? By examining $f$'s output on unbiased off-diagonal samples at test time (middle), one may discover the cue $f$ has adopted. In the example: 1-scale, 2-shape, and 3-color.

for solving a particular task and observe which cues tend to be preferentially adopted by DNNs. The experimental setup is inspired by the Wisconsin Card Sorting Test (WCST, Banno et al. (2012)) in Cognitive Neuroscience[1]. See Figure 1 for an illustration of the setup. It consists of a training set with multiple highly correlated cues (*e.g.* color, shape, and scale) that offer equally plausible pathways to the successful target prediction ($Y \in \{1, 2, 3\}$). We call this a ***diagonal*** training set, highlighting the spatial arrangement of such samples in the product space of all combinations. A model $f$ trained on such a dataset will adopt or neglect certain cues. We analyse the cues adopted by a model by observing its predictions on ***off-diagonal*** samples. With regards to Figure 1, for example, consider the $f$'s prediction for a small, blue triangle as an off-diagonal sample. The prediction values $f(\triangle) \in \{1, 2, 3\}$ tell us which cue the model is biased towards to, *e.g.* if $f(\triangle) = 1$, then f is biased towards scale; if $f(\triangle) = 2$, towards shape; and if $f(\triangle) = 3$ towards color. All three scenarios are plausible and only testing on off-diagonal samples will reveal model bias.

We make important observations on the nature of shortcut bias under `WCST-ML`. We discover that, despite the equal amounts of correlations with the target label, there tends to be a preferential ordering of cues. The preference is largely shared across different DNN architectures, such as feedforward networks, ResNets (He et al., 2015), Vision Transformers (Dosovitskiy et al., 2021), and multiple initial parameters. From the parameter-space perspective, we further observe that the set of solutions $\Theta^p$ biased to the preferred cues takes a far greater volume than those corresponding to the averted cues $\Theta^a$. The loss landscape also tends to be flatter around $\Theta^p$ than around $\Theta^a$.

Why are certain cues preferred to others by general DNNs? We provide an explanation based on the Kolmogorov complexity of cues, which measures the minimal description length for representing cues (Kolmogorov, 1963). Prior studies have shown that in the parameter space of generic DNNs, there are exponentially more Kolmogorov-simple functions than Kolmogorov-complex ones (Valle-Perez et al., 2019; De Palma et al., 2019). Based on these theoretical results, we argue that DNNs are naturally drawn to Kolmogorov-simple cues. We empirically verify that the preferences for cues correlate well with their Kolmogorov complexity estimates.

What are the consequences of the inborn preference for simple cues? Firstly, this may hinder the generalization of DNNs to challenging test scenarios where the simple shortcut cues are no longer valid (Geirhos et al., 2020). Secondly, we expose the possibility that certain protected attributes correspond to the simple shortcut cue for the task at hand, endangering the fairness of DNNs (Barocas et al., 2017). In such a case, human intervention on the learning procedure may be necessary to enforce fairness, for the dataset and DNNs can be naturally drawn to exploit protected attributes.

The primary goal of this manuscript is to shed light on the nature of shortcut biases and the underlying mechanisms behind the scenes. Our contributions are summarized as follows: an experimental setup for studying the shortcut bias in-depth (`WCST-ML`) (§2); novel observations on the nature of shortcut biases, such as the existence of preferential ordering of cues and its connections to the geometry of the loss landscape in the parameter space (§3); an explanation based on the descriptional

---

[1]Original WCST gauges the subjects' cognitive ability to flexibly shift their underlying rules (adopted cues) for categorizing samples. Inability to do so may indicate dysfunctional frontal lobe activities.

complexity of cues (§4); and a discussion on the implications on generalization and fairness (§5), such as the preferential use of ethnical features for face recognition on the UTKFace dataset.

## 2 SETUP

We introduce the setup that will provide the basis for the analysis in this paper. We describe the procedure for building a dataset with multiple equally valid cues for recognition (§2.1). The procedure is applied to DSprites and UTKFace datasets in §2.2. In §2.3, we introduce terminologies for the analysis of the parameter space and make theoretical connections with our data framework.

### 2.1 DATA FRAMEWORK: `WCST-ML`

Many factors affect the preference of models to certain cues. The existence of dominant classes is an example; it encourages models to favor cues conducive to a good performance on the dominant classes (Barocas et al., 2017; Hashimoto et al., 2018). In other cases, some cues have higher degrees of correlation with the target label (Geirhos et al., 2020). In this work, we test whether bias is still present under fair conditions, i.e.: when a training dataset contains a set of valid cues, each of which equally correlates with the targets, will DNNs still have a preference for certain cues? If so, why?

To study this, we introduce a data construction framework called Wisconsin Card Sorting Test for Machine Learners (`WCST-ML`), named after a clinical test in cognitive neuroscience (Banno et al., 2012). See Figure 1 for an overview. As a running example, we assume a dataset where each image can be described by varying two latent variables, object *shape* and object *color*. Let $X$ and $Y$ denote image and label, respectively. We write $X_{ij}$ for the image with color $i$ and shape $j$, where $i, j \in \{1, \cdots, L\}$. When we want to consider $K > 2$ varying factors, we may write $X_{i_1, \cdots, i_K}$ for the image random variable with $k^{\text{th}}$ factor chosen to be $i_k \in \{1, \cdots, L\}$. Importantly, we fix the number of categories for each factor to $L$ to enforce similar conditions for all cues. Similar learning setups have appeared in prior papers: "Cross-bias generalisation" (Bahng et al., 2020), "What if multiple features are predictive?" (Hermann & Lampinen, 2020), and "Zero generalization opportunities" (Eulig et al., 2021). While we fully acknowledge the conceptual similarities, we stress that our work presents the first dedicated study into the cue selection problem and the underlying mechanisms.

The same set of images $\{X_{ij} \,|\, 1 \le i, j \le L\}$ admits two possible tasks: color and shape classification. The task is determined by the labels $Y$. Denoting $Y_{ij}$ as the label for image $X_{ij}$, setting $Y_{ij} = i$ leads to the color classification, and setting $Y_{ij} = j$ leads to the shape classification tasks. We may then build the data distribution for the task at hand via

$$\mathcal{D}_{\text{color}} := \bigcup_{1 \le i,j \le L} (X_{ij}, Y_{ij} = i) \qquad \mathcal{D}_{\text{shape}} := \bigcup_{1 \le i,j \le L} (X_{ij}, Y_{ij} = j) \tag{1}$$

for color and shape recognition tasks, respectively. More generally, we may write

$$\mathcal{D}_k := \bigcup_{1 \le i_1, \cdots, i_K \le L} (X_{i_1, \cdots, i_K}, Y_{i_1, \cdots, i_K} = i_k) \tag{2}$$

for the data distribution where the task is to recognize the $k^{\text{th}}$ cue. We define the **union** of random variables as the balanced mixture: $\bigcup_{i=1}^L Z_i := Z_I$ where $I \sim \text{Unif}\{1, \cdots, L\}$.

We now introduce the notion of a **diagonal dataset**, where every cue (*e.g.* color and shape) contains all the needed information to predict the true label $Y$. That is, a perfect prediction for either color or shape attribute leads to a 100% accuracy for the task at hand. This can be done by letting the factors always vary together $i = j$ in the dataset (and thus the name). We write

$$\mathcal{D}_{\text{diag}} := \bigcup_{1 \le i \le L} (X_{ii}, Y_{ii} = i). \tag{3}$$

Such a dataset completely leaves it to the model to choose the cue for recognition. Given a model $f$ trained on $\mathcal{D}_{\text{diag}}$, we analyse the recognition cue adopted by $f$ by measuring its **unbiased accuracy** on all the cells (See Figure 1). There are $K$ different unbiased accuracies for each task, depending on how the off-diagonal cells are labelled: *e.g.* $\mathcal{D}_{\text{color}}$ and $\mathcal{D}_{\text{shape}}$ in equation 1. For a general setting with $K$ cues, the unbiased accuracy for $k^{\text{th}}$ cue is defined as

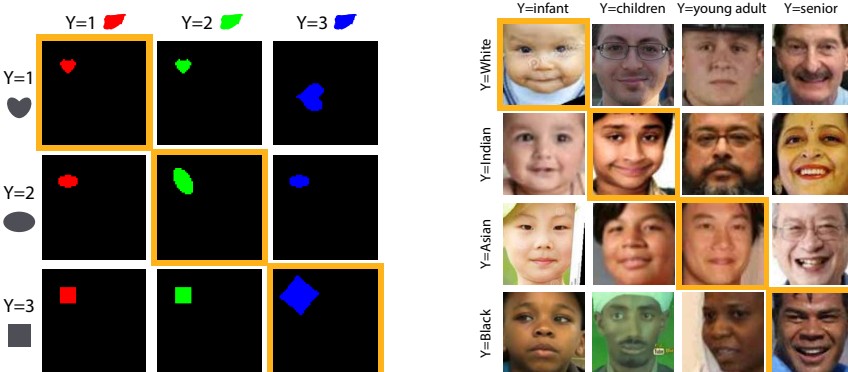

Figure 2: **Dataset samples.** The same set of images for each dataset accommodates two tasks, depending on the labeling scheme (row-wise or column-wise). The diagonal dataset $\mathcal{D}_{\text{diag}}$ (indicated with borders of color ■) is identical regardless of the row- or column-wise labeling. Left: DSprites. Right: UTKFace.

$$\text{acc}_k(f) := \frac{1}{L^k} \sum_{i_1, \cdots, i_K} \Pr\left[f(X_{i_1, \cdots, i_K}) = i_k\right]. \tag{4}$$

**Proposition 1.** *For $k \in \{1, \cdots, K\}$, $\text{acc}_k(f) = 1$ if and only if $f(X_{i_1, \cdots, i_K}) = i_k$ almost surely for all $1 \le i_1, \cdots, i_K \le L$. Moreover, if the condition above holds (i.e. $f$ is perfectly biased to cue k), then $\text{acc}_m(f) = \frac{1}{L}$ for all $m \ne k$.*

The proposition implies that the unbiased accuracy is capable of detecting the bias in a model $f$: $\text{acc}_k(f) = 1$ implies that $f$'s prediction is solely based on the cue $k$. It also emphasizes that it is impossible for a model to be perfectly biased to multiple cues. Finally, we remark that the `WCST-ML` analysis does not require the cues to be orthogonal or interpretable to humans. The only requirement is the availability of the labelled samples $(X_{i_1, \cdots, i_K}, Y_{i_1, \cdots, i_K})$ for the cue of interest.

## 2.2 DATASETS FOR ANALYSIS

**DSprites** (Matthey et al., 2017) is an image dataset of symbolic objects like triangles, squares, and ellipses on black background. It consists of images with **all** possible combinations of five factors: shape, scale, orientation, and X-Y position. The total number of combinations is $3 \times 6 \times 40 \times 32 \times 32 = 737,280$. We augment the dataset with another axis of variation: 4 colors (white, red, green, and blue). This results in $737,280 \times 4 = 2,949,120$ images in the augmented dataset. Each image is of resolution $64 \times 64$. We have identified degenerate orientation labels due to rotational symmetries for certain shapes. We process the data further by collapsing the symmetries to a unified orientation label. For example, we assign the same orientation label for squares at rotations $0$, $\frac{\pi}{2}$, $\pi$, and $\frac{3\pi}{2}$.

**UTKFace** (Zhang et al., 2017) is a face dataset with $33,488$ images with exhaustive annotations of age, gender, and ethnicity per image. We utilize the *Aligned and Cropped Face* version to focus the variation in the facial features only. Ages range from 0 to 116; gender is either male or female; and ethnicity is one of White, Black, Asian, Indian, and Others (e.g. Hispanic, Latino, Middle Eastern) following Zhang et al. (2017). Images are resized from $224 \times 224$ to $64 \times 64$.

**Applying the data framework.** For each analysis, we select a subset of features $\mathbb{S}$ and build the training set $\mathcal{D}_{\text{diag}}$ and test sets $\mathcal{D}_k$ (§2.1) anew based on $\mathbb{S}$. Figure 2 shows examples from DSprites with $\mathbb{S} = \{\text{shape}, \text{color}\}$ and UTKFace with $\mathbb{S} = \{\text{age}, \text{ethnicity}\}$, respectively. To enforce the same number of classes $L$ per feature, we set $l$ as the minimal number of classes among the selected features $\mathbb{S}$. For features with #classes $> L$, we sub-sample the classes to match #classes $= L$. For continuous features like age and rotation, we build categories based on intervals, defined such that the $L$ classes are balanced. We randomly vary cues $\notin \mathbb{S}$ in the constructed datasets $\mathcal{D}_{\text{diag}}$ and $\mathcal{D}_k$. Figure 2 shows the example datasets of $\mathcal{D}_{\text{diag}}$ and $\mathcal{D}_k$ for each dataset.

## 2.3 PARAMETER SPACE AND SOLUTIONS

We define the hypothesis class for DNNs through their weights and biases, collectively written as $\theta \in \Theta$ (*e.g.* all possible weight and bias values of the ResNet50 architecture (He et al., 2016)). We

will use a non-negative loss $\mathcal{L} \geq 0$ to characterize the fit of the model $f_\theta$ on data $\mathcal{D}$. 0-1 loss (and cross-entropy loss) is such an example:

$$\mathcal{L}(\theta; \mathcal{D}) = \Pr_{(X,Y) \sim \mathcal{D}}[f_\theta(X) \neq Y]. \tag{5}$$

We define the **solution set** as $\Theta^\star := \{\theta \in \Theta : \mathcal{L}(\theta; \mathcal{D}) = 0\}$; in practice we define it with a small threshold $\mathcal{L}(\theta; \mathcal{D}) \leq \epsilon$.

We now make a connection between the datasets and solution sets. Assume that we have a dataset of form $(X_{ij}, Y_{ij})$ where $Y_{ij} = i$ (the color task; see §2.1). The corresponding solution set is defined as $\Theta_{ij}^{Y_{ij}} := \{\theta : \mathcal{L}(\theta; (X_{ij}, Y_{ij} = i)) = 0\}$. Define it similarly for the shape task ($Y_{ij} = j$).

**Proposition 2.** *If $\mathcal{D} = \bigcup_{(i,j) \in I}(X_{ij}, Y_{ij})$ for some index set $I$, then the corresponding solution set is the intersection of the solutions sets $\bigcap_{(i,j) \in I} \Theta_{ij}^{\star Y_{ij}}$.*

*Proof.* If $\mathcal{L}$ never attains 0, the solution set is always empty and the statement is vacuously true. Otherwise, we may decompose $\mathcal{L}(\theta)$ into the average $\frac{1}{|I|} \sum_{(i,j) \in I} \mathcal{L}_{ij}(\theta)$. Due to the non-negativity of the loss function, $\mathcal{L}(\theta) = 0$ if and only if $\mathcal{L}_{ij}(\theta) = 0$ for all $(i,j) \in I$. ∎

The proposition leads to a neat summary of the relations among solution sets like $\Theta_{\text{color}} := \{\theta : \mathcal{L}(\theta; (X_{ij}, i)) = 0\}$ and $\Theta_{\text{shape}}$ defined similarly.

**Corollary 3.** $\Theta_{color} \subset \Theta_{diag}$ *and* $\Theta_{shape} \subset \Theta_{diag}$. *Moreover,* $\Theta_{color} \bigcap \Theta_{shape} = \emptyset$.

*Proof.* The first statement immediately follows from the proposition above and the fact that $\mathcal{D}_{\text{diag}} = \mathcal{D}_{\text{color}} \bigcap \mathcal{D}_{\text{shape}}$. For the second, observe that there is no function that satisfies $f_\theta(X_{ij}) = i$ and $f_\theta(X_{ij}) = j$ at the same time for $i \neq j$. ∎

We may easily extend the result to datasets with $K$ cues. Given the solutions sets $\Theta_k$ for each cue $1 \leq k \leq K$, we have $\Theta_k \subset \Theta_{\text{diag}}$ for all $k$ and $\{\Theta_k\}_k$ are pairwise disjoint (diagram on the right). This follows from Proposition 1 that a single model cannot achieve low error for multiple cues ($\mathcal{D}_k$) at the same time.

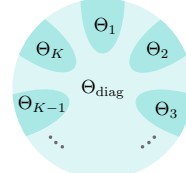

## 3 OBSERVATIONS

Based on the proposed WCST-ML, we study the natural tendencies of deep neural networks (DNNs) to favor certain cues over the others. We also present a parameter-space view that hints to the underlying mechanisms for the cue preference.

### 3.1 PREFERENTIAL ORDERING OF CUES

In the first set of experiments, we observe which cues are preferentially used by the network when training on a dataset composed of an equally represented set of cues: the *diagonal dataset* $\mathcal{D}_{\text{diag}}$ in §2.1. We train three qualitatively different types of DNNs: (1) Feed Forward neural network (FFnet), (2) ResNet with depth 20 (ResNet20), and (3) Vision Transformer (ViT). Details of the architectures and training setups can be found in Appendix §A. The results are summarized in Figure 3.

**Models adopt cues with uneven likelihood.** Figure 3 shows the unbiased accuracies of the models when trained on the diagonal sets for each dataset (DSprites and UTKFace) with the involved cues denoted as $\mathbb{S}$. We report the diagonal accuracy and the $K$ unbiased accuracies for each cue in $\mathbb{S}$ on the held-out test set. The number of labels $L$ is 3 for DSprites and 2 for UTKFace. On DSprites, we observe that for all architectures, color is by far the most preferred cue (unbiased accuracy near 100%). For FFnet and ViT, the extreme color bias forces the accuracy on other cues to be $\frac{1}{L} = 33.3\%$ as predicted by Proposition 1. ResNet20 behaves less extremely and picks up scale, shape, and orientation, in the order of preference. On UTKFace, there is no extreme dominant cue, but there is a general preference to use ethnicity cues for making predictions, followed by gender and age cues. The trend is clear for all models, and while the FFnet model shows a slightly more variable performance across runs, the ranking of cues is preserved throughout. The preference of

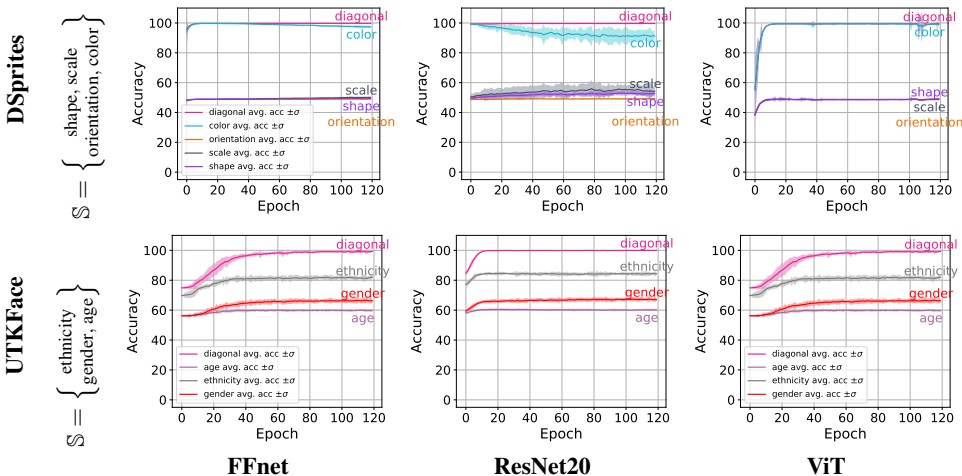

Figure 3: **Preferential ordering of cues.** Diagonal and unbiased accuracies of models trained on the diagonal training set $\mathcal{D}_{\mathrm{diag}}$ composed of the cues defined by $\mathbb{S}$ and tested on the off-diagonal sets $\mathcal{D}_k$ where $k \in \mathbb{S}$.

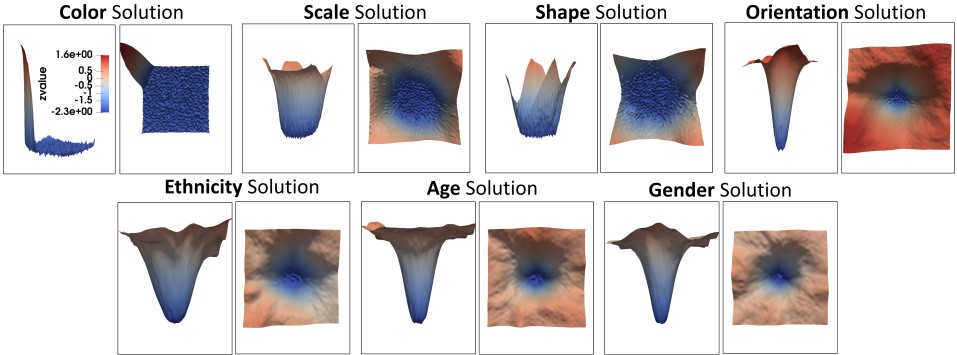

Figure 4: **3-D visualisation of loss surface around different solution types $\Theta^{\star}$.**
For each cue, we show 3-D plots of the same loss surface (side and top views). $X$ and $Y$ axes are randomly chosen directions in the parameter space; $Z$ axis shows the loss value at each $(X, Y)$ measured against the diagonal data $\mathcal{D}_{\mathrm{diag}}$. All the surfaces are shown at the same scale.

ethnicity echoes the DNN's preference of color as a strong cue in DSprites; *DNNs are drawn to utilizing skin color.* Another curious phenomenon is that none of the three cues sufficiently explains the 100% diagonal accuracy, suggesting the existence of an unlabeled shortcut cue other than the three considered. Finally, when the most dominant cue is left out, other cues activate in its place: scale for DSprites and gender for UTKFace (Appendix §B).

**Consistency of preferences.** We have observed a consistent preferential ordering of cues shown by qualitatively different types of architectures. The results suggest the existence of a common denominator that depends solely on the nature of the cues, rather than the architectures. We finally remark that the experiments were run 10 times with different random initialization; error bars in Figure 3 show $\pm\sigma$ standard deviation. The small variability observed across most training shows the consistency of the preferential ranking with respect to initialization. For a pair of cues, we refer to the one that is more likely to be chosen by a model trained on $\mathcal{D}_{\mathrm{diag}}$ as the **preferred cue** and the other as the **averted cue**.

### 3.2 THE ABUNDANCE OF PREFERRED-CUE SOLUTIONS

Why are certain cues preferred to others? We look for the answers in the parameter space. More specifically, we study the geometric properties of the loss function and the corresponding solution sets in the parameter space. The section is based on §2.3. We write the solution set corresponding to the preferred cues as **preferred solutions** $\Theta_p$ and averted cues as **averted solutions** $\Theta_a$.

**How to find the averted solutions $\theta \in \Theta_a$?**    Naively training a DNN on $\mathcal{D}_{\text{diag}}$ will most likely result in a preferred solution $\Theta_p$. We thus use the full dataset $\mathcal{D}_k$ corresponding to the averted cue $k$. (§2.1) as the training set to find an averted solution $\theta \in \Theta_a$. To ensure a fair comparison in terms of the amount of training data, we also use the full dataset $\mathcal{D}_k$ for the preferred cues.

**Qualitative view on the loss surface.**    We examine the geometric properties of the loss surface for ResNet20 around different solutions types (preferred and averted). We use the 2D visualization tool for loss landscape by Li et al. (2018). The X-Y grid is spanned by two random vectors in $\mathbb{R}^D$ where $D$ is the number of parameters. In Figure 4, we observe that the preferred solutions (*e.g.* $\Theta_{\text{color}}$) are characterized with a flatter and wider surface in the vicinity. In contrast, local surface around averted (*e.g.* $\Theta_{\text{orientation}}$) solutions exhibit relatively sharper and narrower valleys. A similar trend is observed for the UTKFace cues at a milder degree: $\Theta_{\text{ethnicity}}$ is a slightly flatter solution than $\Theta_{\text{age}}$.

**Size of the basin of attraction.**    We compare the **size** of the preferred $\Theta^p$ and averted $\Theta^a$ solution sets using the notion of the **basin of attraction (BA) for a cue** $k$: a subset $N$ in the parameter space such that initializing the training with a point $\theta_0 \in N$ leads to a $k$-biased solution $\theta^\star \in \Theta_k$. To measure the **size** of the BA for $\Theta_k$, we start by obtaining a solution $\theta_k \in \Theta_k$ biased to the cue $k$. 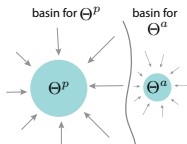
We then check when the final models trained from perturbed points $\theta_k + v$ (where $\|v\|_2$ is small) stops being biased to the cue $k$. We treat the norm of the bias-shifting critical perturbation $\|v\|_2$ as the size of the BA for $\Theta_k$.

**Preferred solutions have wider BA.**    Figure 5 shows the size of BA measured according to the method above. The upper row shows initialization around solutions biased to cues in DSprites. For the color cue, we observe that initializing far away from the color-biased solution still results in a color-biased solution. For shape, scale, and orientation cues, we observe that initializing away from the respective solutions gradually results in color-biased solutions with a reduction in the bias towards the respective biases in the unperturbed initial parameters. The sizes of basins of attraction are ordered according to color>shape>scale>orientation, largely following the preferential ordering observed in §3.1. The lower row of Figure 5 shows the results for the three cues in UTK-Face. As opposed to DSprites, the solutions for ethnicity, age, and gender do not re-converge to a clear cue after retraining for only 50 epochs. The sizes of basins of attraction are ordered as ethnicity>gender>age, following the preferential ordering of cues observed in §3.1.

**Path connectivity.**    Recent works have proposed methods to find near zero-loss paths between two solutions (Garipov et al., 2018; Draxler et al., 2018; Benton et al., 2021). We extend the technique to find the zero-loss (w.r.t. $\mathcal{D}_{\text{diag}}$) path between preferred and averted solutions. We locate the flipping moment for the bias along the path; there must be a *bias-shift boundary* where the bias to the preferred cue shifts to the averted cue along the path (by an intermediate- 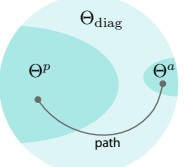
value-theorem-like argument). By comparing the length of the segment up to this boundary, we may estimate the relative abundance of the preferred and averted solutions in the parameter space.

**Paths connecting preferred and averted solutions and the bias-shift boundary.**    Figure 6 visualizes the paths between the preferred and averted solutions as well as the corresponding shifts in biases along the path. We observe that the paths generally show a single shift in bias. The shift is also usually quick; for the most part of the path, the solutions tend to be heavily biased towards either of the cues. For the path from $\Theta_{\text{color}}$ to $\Theta_{\text{orientation}}$, we observe a tendency for $\Theta_{\text{color}}$ to dominate the path until near $\Theta_{\text{orientation}}$, suggesting the relative abundance of color-biased solutions $\Theta_{\text{color}}$ in the set of diagonal solutions $\Theta_{\text{diag}}$ compared to the orientation $\Theta_{\text{orientation}}$. The path from $\Theta_{\text{ethnicity}}$ to $\Theta_{\text{gender}}$ is composed of a longer segment for the $\Theta_{\text{ethnicity}}$ part. The relative abundance of cues in the diagonal solutions $\Theta_{\text{diag}}$ resonates the preferential ordering of cues seen in §3.1.

## 4   EXPLANATIONS

We have observed intriguing properties of cues and shortcut biases. Even under the equal opportunities for the cues to be adopted, models have a preference for a few cues over the others. The

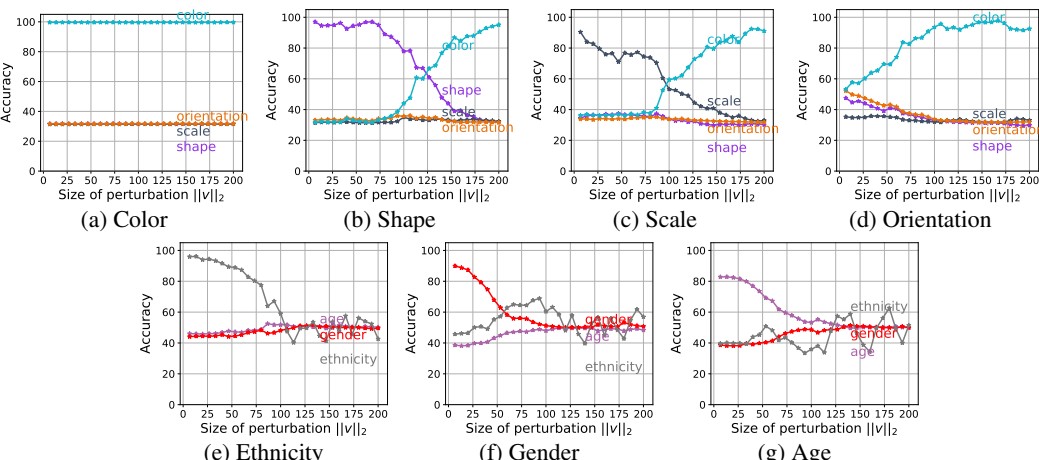

Figure 5: **Basin of attraction.** We initialize ResNet20 around solutions biased to each of the four cues in DSprites (upper row) and three cues in UTKFace (lower row). We report the averaged unbiased accuracies for models initialized with perturbed solutions $\theta^\star + v$. The x-axes are the perturbation sizes $\|v\|_2$. From the perturbed initial parameters, each model is trained for 50 epochs.

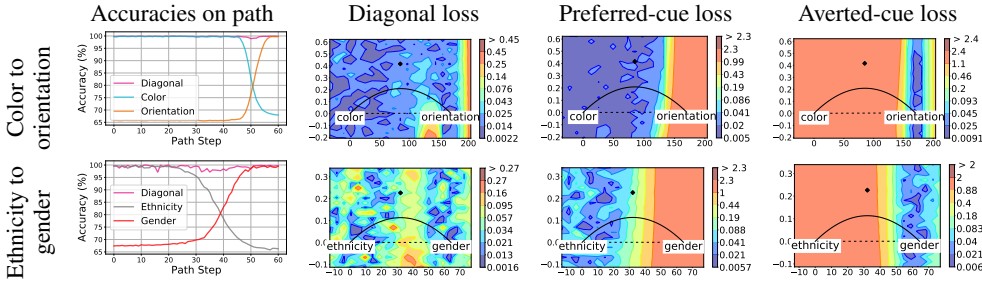

Figure 6: **Paths connecting $\Theta^p$ and $\Theta^a$.** We compute paths between preferred and averted solutions along which the diagonal-dataset ($\mathcal{D}_{\mathrm{diag}}$) loss is zero. On the leftmost column, we show the unbiased accuracies for the preferred and averted cues along the path; we observe a bias shift in the middle in each case. The three rightmost columns visualize the loss values around the zero-loss paths (solid curves), with respect to the losses computed on diagonal $\mathcal{D}_{\mathrm{diag}}$, preferred cue $\mathcal{D}_p$, and averted cue $\mathcal{D}_a$ datasets. The zero-loss paths are quadratic Bezier curves, parametrised by three dots shown in the plots: the two end points (preferred and averted solutions) and a third point that regulate the curve shape by "pulling" it. Inspired by Garipov et al. (2018).

preference is stable across architectures and weight initialization. We have also found a strong correlation between the preference for a cue and the abundance of solutions biased to that cue. In this section, we explain the observations based on the *complexity of cues*.

**Kolmogorov complexity of cues (KCC).** We formally define **cue** as the condition distribution $p_{Y|X}$ for a given input distribution $p_X$; in a sense, it concurs with the definition of *task*. Kolmogorov complexity (KC, Kolmogorov (1963)) measures complexity of binary strings based on the minimal length among programs that generate the string. Achille et al. (2018) have defined the **Kolmogorov complexity of a task** (or **a cue**) $p_{Y|X}$ as:

$$K(p_{Y|X}) = \min_{\mathcal{L}(f;X,Y)<\delta} K(f) \tag{6}$$

for some sufficiently small scalar $\delta > 0$. $f$ denotes a discriminative model mapping $X$ to $Y$ and $\mathcal{L}$ is a suitable loss function. $K(f)$ is the KC for model $f$. Intuitively, $K(p_{Y|X})$ measures the minimal complexity of the function $f$ required to *memorize* the labeling $p_{Y|X}$ on the training set (*i.e.* $\mathcal{L} < \delta$).

**Measuring KCC.** We estimate the KC of four cues (color, scale, shape, and orientation) considered in DSprites based on equation 6. Since it is impractical to optimise $K(f)$ over an open-world of *all* possible models $f$, we constrain the search to a *homogeneous* family of models $f \in \mathcal{F}$. We use ResNet20 with variable channel sizes (Zagoruyko & Komodakis, 2016) for $\mathcal{F}$. Given that KC is uncomputable, we use the number of parameters in $f$ as an approximation to $K(f)$ in practice

(Valle-Perez et al., 2019). We set the memorization criterion $\mathcal{L}(f; X, Y) < \delta$ as "training-set error for $f$ is $< 1\%$". Under this setup, we have verified that the complexities of cues are 1.2K parameters for color, 4.6K for scale, 17.6K for shape, and $> 273$K for orientation. We were not able to fully memorize the orientation labels under the network scales we considered. The complexity ranking strongly concurs with the preferential ordering found in §3.1.

**The simplicity bias in DNN parameter space.** How are the cue complexity and shortcut bias connected? We explain the underlying mechanism based on a related theory in a different context: explaining the generalizability of DNNs with the *simplicity bias* (Valle-Perez et al., 2019; Shah et al., 2020). It has been argued that, despite a large number of parameters, DNNs still generalize due to the inborn tendency to encode simple functions. This phenomenon has filled the theoretical gap left by the classical learning theory. The argument is that a (uniform) random selection of parameters for a DNN is likely to result in a function $f$ that is simple in the sense of KC. More precisely, the likelihood of a function $f$ being encoded by a DNN is governed by the inequality $p(f) \lesssim 2^{-a\widetilde{K}(f)+b}$, where $K(f)$ is the KC of $f$ and $a > 0$, $b$ are constants independent of $f$ (Valle-Perez et al., 2019). This inequality stems from the Algorithmic Information Theory and has been validated to hold for a large variety of systems that arise naturally, such as RNA sequences and solutions to a large class of differential equations (Dingle et al., 2018). The argument concludes that the natural abundance of simple functions in the parameter space makes it more likely for the solutions to be simpler (Wu et al., 2017), leading to the *simplicity bias* and thus helping DNNs generalize well.

**The simplicity bias leads to shortcuts.** We have seen that cues have different KCs, necessitating DNNs with different complexities to fully represent them. The simplicity bias in the parameter space explains the abundance of solutions biased to simple cues like color $\Theta_{\text{color}}$ rather than complex cues like orientation $\Theta_{\text{orientation}}$ (§3.2). The difference in the abundance in turn makes it more likely for a model trained on the diagonal dataset $\mathcal{D}_{\text{diag}}$ to be biased towards simple cues (§3.1).

## 5 DISCUSSION AND CONCLUSION

We have delved into the shortcut learning phenomenon in deep neural networks (DNNs). We have devised a training setup `WCST-ML` where multiple cues are equally valid for solving the task at hand and have verified that, despite the equal validity, DNNs tend to choose cues in a certain order (*e.g.* color is preferred to orientation). We have shown that the simple cues (in the sense of Kolmogorov) are far more frequently represented in the parameter space and thus are far more likely to be adopted by DNNs. We conclude the paper with two final remarks on the implications of our studies.

**Simplicity bias and generalization.** The simplicity bias theory (Valle-Perez et al., 2019) has been developed to explain the unreasonably good generalizability of DNNs. Our paper shows that it may interfere with the generalization. This apparent paradox stems from the difference in the setup. The simplicity bias is often safe to be exploited in the standard iid setup, albeit with caveats argued by Shah et al. (2020). On the other hand, it may lead to less effective generalization across distribution shifts as in our learning setup (Geirhos et al., 2020), if the simple cues are no longer generalizable. Thus, for the generalization across distribution shifts, machine learners may need to be given additional guidance to overcome the inborn tendencies to pick up simple cues for recognition.

**Societal implications.** An issue with the simplicity bias and shortcut learning is that relying on simple cues is sometimes unethical. We have seen in the UTKFace experiments that *ethnicity* tends to be a more convenient shortcut than *age*, for example, if they equally correlate with the target label at hand. This is an alarming phenomenon. It emphasizes the importance and inevitability of active human intervention to undo certain naturally-arising biases to socially harmful cues in learned models. It also confirms that we are on the right track to actively regularize the models to behave according to the social norms (Barocas et al., 2017; Mehrabi et al., 2021).

**Toolbox contribution.** Along the way, we contribute interesting novel tools that future research can utilise, such as the computation of Kolmogorov complexity (KC) for generic cues (equation 6) and the method for finding a heterogeneous zero-loss curve connecting two solutions biased to different cues (§3.2; Garipov et al. (2018)).

REPRODUCIBILITY

We understand and resonate with the importance of reproducible research practices in machine learning. The reproducibility of this paper hinges on our use of open-source data, models, training techniques, and deep learning frameworks. We have specified a detailed experimental setup in §2.2 and §A. For the consistency of experimental observations, we have run the experiments multiple times with different random seeds, whenever applicable. For those cases, we have reported the mean and standard deviation for each data point.

ETHICS STATEMENT

We delve into the potential ethical issues with DNNs. Our work excavates the potential ethical harms caused by deep neural networks' natural tendencies to prefer simple cues, where using the simple cues may not conform well to the social norm (*e.g.* ethnicity cue). We have thus advocated an active human involvement to regularize such DNNs to adopt more socially acceptable cues. As for the experiments, part of our studies is based on a face dataset (UTKFace). We confirm that the UTKFace dataset has been released with an open-source license with the non-commercial clause[2]. The copyright for the images still belongs to their respective owners. Inheriting the policy of UTKFace, we will erase the face samples in Figure 2 upon request by the owners of the respective images.

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

## A  MODEL ARCHITECTURES

**FFnet.**  The FFnet architecture tested is a fully connected feedforward neural network with 5 hidden layers with respectively 4096, 2048, 600, 300 and 100 hidden units. ReLU activations are used throughout. We train it using the Adam optimizer with the default parameters given by PyTorch 1.8.0.

**ResNet20.**  The ResNet20 architecture is adapted from He et al. (2016), with a depth of 20. We train it using the Adam optimizer with the default parameters given by PyTorch 1.8.0.

**ViT.**  The Visual Transformer architecture is adapted from (Touvron et al., 2021). Given the dimensions of the images in ColorDSprites and consequently UTKFace, we implement a ViT model with patch size 8, embedding dimension 192, depth 12, and 3 attention heads. We train the model with the SGD optimizer with learning rate $5 \times 10^{-3}$, momentum 0.9, and weight decay $1 \times 10^{-4}$.

## B  PREFERENTIAL ORDERING AFTER REMOVING THE DOMINANT CUE

For both DSprites and UTKFace, removing the dominant cue (color and ethnicity respectively) from $\mathbb{S}$ does not change the ranking of the remaining cues; *e.g.* scale and gender then become the dominant features in respective datasets. See Figure B.1 that extends the main paper Figure 3.

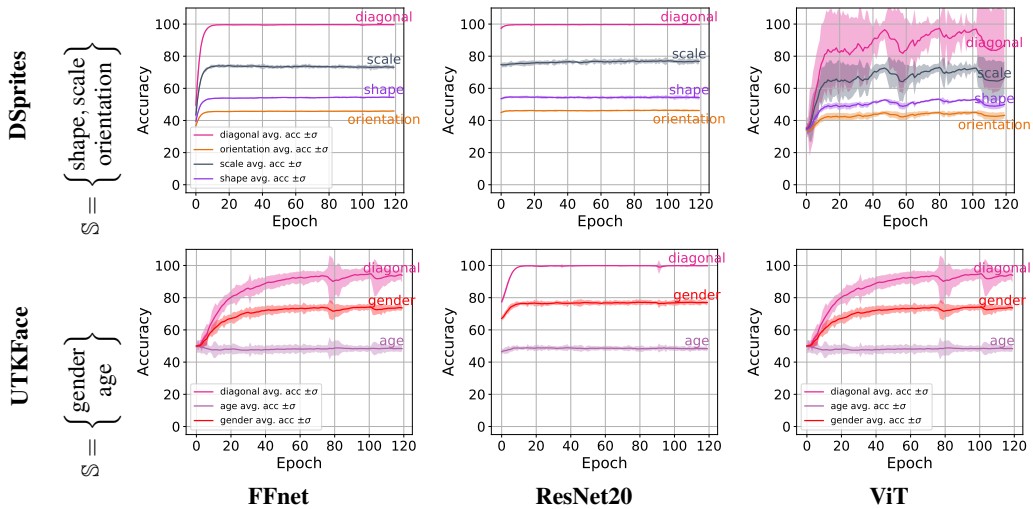

Figure B.1: **Preferential ordering of cues.** Diagonal and unbiased accuracies of models trained on the diagonal training set $\mathcal{D}_{\text{diag}}$ composed of the cues defined by $\mathbb{S}$ and tested on the off-diagonal sets $\mathcal{D}_k$ where $k \in \mathbb{S}$.

## C  LOSS CURVATURE AROUND SOLUTIONS

We supplement the visualization of loss landscape around solutions in Figure 4 of main paper with quantitative estimation of the respective curvatures. The mean curvature of a surface defined by function $f$ is defined as:

$$\kappa(f) := \frac{1}{D}\Delta f := \frac{1}{D}\left(\frac{\partial^2 f}{\partial x_1^2} + \cdots + \frac{\partial^2 f}{\partial x_D^2}\right) \tag{C.1}$$

To make it computationally tractable for large $D$ (*e.g.* DNN parameters), we estimate the curvature with a Monte-Carlo estimation as done in Izmailov et al. (2018). For a point $x \in \mathbb{R}^D$, a sufficiently small scalar $\epsilon > 0$, and a unit sphere $S^{D-1} \subset \mathbb{R}^D$, the curvature is computed by:

$$\kappa(f, x, \epsilon) \approx \frac{1}{\epsilon^2}\left[\mathbb{E}f(x + \epsilon r) - f(x)\right] \quad \text{where} \quad r \sim \text{Unif}(S^{D-1}). \tag{C.2}$$

How does this approximate the mean curvature? We provide the following proposition.

**Proposition 4.** *For a point $x \in \mathbb{R}^D$, a scalar $\epsilon > 0$, a twice-differentiable function $f$, and a unit sphere $S^{D-1} \subset \mathbb{R}^D$, the following quantity*

$$\frac{1}{\epsilon^2} \left[ \mathbb{E}f(x + \epsilon r) - f(x) \right] \quad where \quad r \sim Unif(S^D) \tag{C.3}$$

*approximates the Laplacian $\Delta f$ up to a constant multiplier.*

*Proof.* We begin with a Taylor expansion on $f(x + \epsilon r)$:

$$f(x + \epsilon r) = f(x) + \epsilon r^T \nabla f(x) + \frac{1}{2}\epsilon^2 r^T \nabla^2 f(x) r + h(\epsilon, r) \tag{C.4}$$

where $\frac{h(\epsilon, r)}{\epsilon^2} \to 0$ as $\epsilon \downarrow 0$ for any $r$. Thus,

$$\frac{1}{\epsilon^2} \left[ \mathbb{E}f(x + \epsilon r) - f(x) \right] = \frac{1}{2}\mathbb{E}\left[ r^T \nabla^2 f(x) r + \frac{h(\epsilon, r)}{\epsilon^2} \right] \tag{C.5}$$

$$= \frac{1}{2}\mathbb{E}[r^T \nabla^2 f(x) r] + o(\epsilon) \tag{C.6}$$

by $\mathbb{E}r = 0$ and the Dominated Convergence Theorem for Lebesgue integrals (also noting that $r$ is defined on a compact set $S^{D-1}$).

We now apply a change of basis such that the Hessian $\nabla^2 f(x)$ is diagonalised: $\nabla^2 f(x) = Q^T \Lambda Q$. This is possible because the Hessian is a real, symmetric matrix. Since the change of basis do not alter the distribution for $r$, we compute

$$\mathbb{E}[r^T \nabla^2 f(x) r] = \mathbb{E}[r^T \Lambda r] = \text{Var}(r_1)\lambda_1 + \cdots + \text{Var}(r_D)\lambda_D \tag{C.7}$$

$$= \text{Var}(r_1)(\lambda_1 + \cdots + \lambda_D) = \text{Var}(r_1)\text{tr}(\nabla^2 f(x)) = \text{Var}(r_1)\Delta f(x) \tag{C.8}$$

using the isotropicity of $r$ and the preservation of the trace against changes of basis. We thus conclude

$$\frac{1}{\epsilon^2} \left[ \mathbb{E}f(x + \epsilon r) - f(x) \right] = C\Delta f(x) + o(\epsilon) \tag{C.9}$$

where $C$ is a constant independent of $f$ and $x$. ∎

**Results.** Figure C.1 shows the approximated curvature at varying radius values $\epsilon$ around different solutions for ResNet20. We have used 100 Monte-Carlo samples. We observe that for DSprites, the curvature of the orientation loss surface tends to overshoot quickly as one walks away from the solution point. Surfaces for other cues tend to be smoother. On UTKFace, gender tends to provide a steeper surface than the other cues like age and ethnicity.

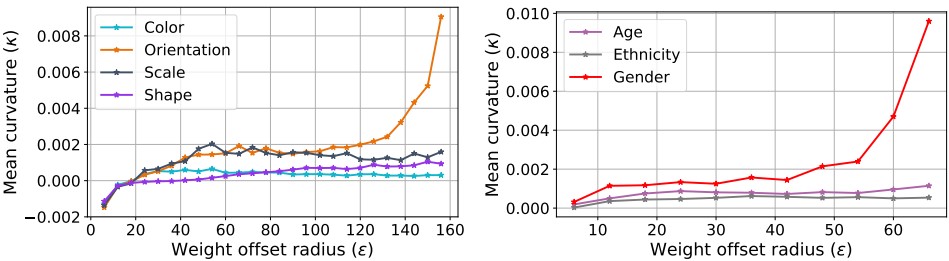

Figure C.1: **Mean curvature around different solutions.** Left: DSprites. Right: UTKFace.

