# OpenReview forum: "Which Shortcut Cues Will DNNs Choose? A Study from the Parameter-Space Perspective"
_ICLR.cc/2022/Conference — ICLR 2022 Poster_

### Official Review · Reviewer_QJQp · 2021-11-02

**Correctness:** 4
**Technical Novelty And Significance:** 3
**Empirical Novelty And Significance:** 2
**Recommendation:** 5
**Confidence:** 4

**Main Review:**

The paper is written well, the claims are laid out clearly, and the method is largely well described. The approach seems methodically sound given the problem statement  (caveat in W2 below). The path of investigation is fundamentally valuable in the sense that models of biases and generalization of networks are helpful tools.

Weaknesses:
[W1] Some parts of the exposition need minor clarification:
(a) p6: "The trend is clear for ResNet20 and ViT, while the ethnicity and gender preferences are within the error bars for FFnet." This refers to Figure 3, and I cannot understand this from Figure 3, in the least it is confusingly stated. The ethnicity and gender graphs for FFNet in figure 3 seem separated by several standard deviations according to the error bars.
(b) Figure 6, three right columns: What does the dot and the black arch signify? I was not able to gain this from text or caption. I assume that the zero loss path is the dashed line?

[W2] A somewhat methodical weakness of the paper seems to be the problem statement. For instance in the introductory example of section one/figure 1, the authors imply that the model is biased towards scale (then shape, then color). I do agree that this is consistent with the model preferring the property of scale in making the decision. Is this bias, though? In order to evidence bias, I'd need a truth/label assignment on the off diagonals (e.g. indicating that blue small triangle is not supposed to be class 1). Without this truth assignment I do not see the bias of the model.

[W3] The main weakness of the paper seems to me that I am asking myself: What have I learned after reading the paper? I feel that the experiment setup is sound in the stated sense: what does the network learn if it only ever sees pathological samples (i.e. where all latent variables are exactly correlated). However, the presented answers that networks tend to learn simpler cues, where simpler can be measured by Kolmogorov complexity are not overly surprising or counter-intuitive. Perhaps the authors can rephrase the contributions more clearly towards the benefits? For instance: With the tools, analysis and exposure of this paper, what can I now see different, do different, etc. What are possible next steps where stronger impact is on the horizon?

**Summary Of The Paper:**

The paper discusses biases in inductive learning in deep neural networks that stem from pathologically sampled data. The authors pose a problem set where a trainer only sees samples where two or more latent values can only be observed in a fully correlated fashion (e.g. scale, color, shape). They then design various criteria and protocols to gain insight in the behavior of the learned model when the correlation does not hold any more (i.e. samples that have not been seen in the training data). They conclude that in this case of generalization to unseen data, the trained model has an implicit bias towards choosing (1) mostly single cues to determine the predicted label (e.g. color only), (2) that more preferred cues are simpler than less preferred cues and (3) that the underlying solution space of possible parameters has more solutions that prefer the simple cues. They demonstrate empirical results on variations of existing datasets (DSprites, UTKFace).

**Summary Of The Review:**

The paper addresses an important topic, but I feel that the overall impact, measured by depth of presented contributions is too shallow.

---

> ### Author Response · Authors · 2021-11-17
> **Response 2/2**
>
> > [W3] The main weakness of the paper seems to me that I am asking myself: What have I learned after reading the paper? I feel that the experiment setup is sound in the stated sense: what does the network learn if it only ever sees pathological samples (i.e. where all latent variables are exactly correlated). However, the presented answers that networks tend to learn simpler cues, where simpler can be measured by Kolmogorov complexity are not overly surprising or counter-intuitive. Perhaps the authors can rephrase the contributions more clearly towards the benefits? For instance: With the tools, analysis and exposure of this paper, what can I now see different, do different, etc. What are possible next steps where stronger impact is on the horizon?
>
> Many thanks for this comment. The reviewer’s concerns are well received, and we take it upon ourselves to improve the discussion and conclusion section to better explain the practical benefits deriving from the experiments in this work.
>
> The current form of the paper is focused more on conceptual contributions. As the reviewer has summarized well, we show that shortcut learning may not only stem from the disparities in cues’ statistical correlations to the target task but also from their very nature: cues have different inherent Kolmogorov complexities (KC). An interesting implication of our conceptual contribution is that our conclusion goes against the common wisdom in machine learning: given simple and complex solutions, it is better to choose the simple one (Occam’s razor). There are common DNN-training recipes designed to encourage simpler solutions, such as stochastic gradient descent, dropout, stochastic depth, and weight decay. An immediate future work would be to re-assess their utilities when applied to de-biasing or fairness scenarios, as they may encourage unwanted shortcuts.
>
> The less pronounced yet crucial contribution of this paper is the toolbox for analysis. We have introduced the measurement of KC for generic cues, which will correlate well with the models’ preference for the cues. There is nothing that stops us from computing the KC even for real-world cues; the only requirement is access to samples labelled according to the cue of interest. The estimated KCs may be useful for many downstream applications. For example, one may optimize the needed amount of the off-diagonal samples to de-bias a model based on the KC estimates. This will be highly useful in practice for reducing the data collection costs; collecting or synthesising the off-diagonal samples is typically expensive as they would seldom appear in nature (e.g. trucks on lakes).
>
> The combination of the mode-connectivity technique [1] and the bias analysis technique in our paper provides further avenues for future research directions. In our paper, we have used the mode-connectivity technique to find a path from a preferred solution to an averted one along which the original-task loss (i.e. diagonal loss) is effectively zero. A particularly successful application of the mode-connectivity technique has been the ensemble of models along the path to improving generalisation [1]. Such a path ensemble tends to generate a more diverse set of models than locally perturbing the solutions [1]. We can extend this idea further in our setup where the endpoints of the path correspond to solutions attending to different cues. Samples along such a heterogeneous curve are likely to be much more diverse and could result in a more effective ensemble. We may also control the type and amount of bias in the resulting ensemble by adjusting the concentration of sampling probabilities along the curve.
>
> Above are only a subset of possible research directions that our analyses and tools may inspire. We have revised the manuscript to aid readers in quickly grasping the contributions and gaining insights like the above. Please take a look at the added “toolbox contribution” paragraph in Section 5 that summarises the above possibilities. Let us know if the reviewer believes further improvement will be necessary - we will be happy to engage in further discussions and revisions.
>
> [1] Garipov et al. Loss surfaces, mode connectivity, and fast ensembling of dnns. In NeurIPS 2018.

---

> > ### Comment · Reviewer_QJQp · 2021-11-29
> > **Response**
> >
> > I thank the authors for their thorough consideration of my review. While the authors responses and updated draft definitely point in the right direction, I remain of the opinion that the current version is not quite ready for exposure to a larger audience on the main track. I have upgraded my rating of technical novelty from 2 to 3 to recognize the components the authors point out in the response: (a) toolbox for KC estimation and zero-loss path finding, (b) added outlook on the thinking around mode-connectivity for potential impact in ensemble models.
> >
> > In case of final rejection of this submissions, I'd suggest to the authors to strengthen a resubmission by going deeper on the empirical side on questions they have raised in the rebuttal, e.g.
> >
> > "We can extend this idea further in our setup where the endpoints of the path correspond to solutions attending to different cues. Samples along such a heterogeneous curve are likely to be much more diverse and could result in a more effective ensemble. We may also control the type and amount of bias in the resulting ensemble by adjusting the concentration of sampling probabilities along the curve."
> >
> > I feel that convincing empirical results from the above in a practical, "reasonably real-world" case would be very well received by the parts of the community that are concerned with generalization and robustness.

---

> ### Author Response · Authors · 2021-11-17
> **Response 1/2**
>
> We thank the reviewer for the thorough review and insightful comments. We are glad the reviewer sees fundamental value in this path of investigation. We lay below our response to the main comments in the review.
>
> >  [W1]: Some parts of the exposition need minor clarification:
>
> >  (a) p6: "The trend is clear for ResNet20 and ViT, while the ethnicity and gender preferences are within the error bars for FFnet." This refers to Figure 3, and I cannot understand this from Figure 3, in the least it is confusingly stated. The ethnicity and gender graphs for FFNet in figure 3 seem separated by several standard deviations according to the error bars.
>
> As mentioned by the reviewer, the description referring to the trends of the FFNet model in Figure 3 was misleading and was referring to an older version of the figure. The up-to-date figure shows indeed a clear trend for all models. While the FFNet model shows a more variable performance across runs, the ranking of cues is preserved throughout. The statement has been updated accordingly in the manuscript (Section 3.1, under paragraph “Models adopt cues with uneven likelihood”).
>
> > (b) Figure 6, three right columns: What does the dot and the black arch signify? I was not able to gain this from text or caption. I assume that the zero loss path is the dashed line?
>
> The three rightmost columns in Figure 6 are mainly based on Figure 1 of [1]. [1] has found that it is almost always possible to find a zero-loss **quadratic curve** between two solutions, while finding a zero-loss linear path is not as easy. A quadratic curve (more precisely a Bezier curve [1]) is parameterized with three points: two end points and the third point that determines the shape of the curve. In our plots, we have shown all three points as black dots, the zero-loss quadratic curve as the solid line, and the linear segment connecting the solutions as a dashed line (which does not guarantee zero loss). We have improved the explanation in the caption of Figure 6.
>
> [1] Garipov et al. Loss surfaces, mode connectivity, and fast ensembling of dnns. NeurIPS 2018.
>
> > [W2] A somewhat methodical weakness of the paper seems to be the problem statement. For instance in the introductory example of section one/figure 1, the authors imply that the model is biased towards scale (then shape, then color). I do agree that this is consistent with the model preferring the property of scale in making the decision. Is this bias, though? In order to evidence bias, I'd need a truth/label assignment on the off diagonals (e.g. indicating that blue small triangle is not supposed to be class 1). Without this truth assignment I do not see the bias of the model.
>
> We believe the description for Figure 1 and the related text have not been exceptionally clear in showcasing the focal takeaways in WCST-ML, and could have led to misinterpretations of the Figure. In Figure 1, we show objects which vary in shape, scale and color. In the diagonal examples, these object cues are fully correlated, thus we have a small-red-circle (label 1), a medium-green-triangle  (label 2), and a large-blue-square (label 3). In the lower right corner of the figure, instead of implying which cue the example model is biased towards, we instead only show different possible scenarios:
> - If we observe that $f$(small-blue-triangle) = 1, then we can infer that $f$ is biased to scale.
> - If $f$(small-blue-triangle) = 2, then $f$ is biased to shape.
> - If $f$(small-blue-triangle) = 3, then $f$ is biased to color.
>
> This is to illustrate how, by experiment design, we can determine which cue the model $f$ has been biased towards by looking at the off-diagonal samples. As the reviewer said, it is possible to reveal the bias for $f$ **if and only if** you have access to off-diagonal samples. If such off-diagonal samples are furthermore labelled, then we can even make a judgment on the model, for example on whether the model is biased towards helpful cues or wrong cues.
>
> Thanks to this comment we have revised the caption for Figure 1 and the related paragraph to clarify these important aspects and improve the overall presentation of WCST-ML.

---

### Official Review · Reviewer_s3vA · 2021-11-02

**Correctness:** 3
**Technical Novelty And Significance:** 3
**Empirical Novelty And Significance:** 3
**Recommendation:** 8
**Confidence:** 4

**Main Review:**

The paper is a very interesting read. It provides an interesting analysis of model preferences for various visual cues. The analysis shows the preference of visual models towards low complexity visual cues, such as color and ethnicity.

Main comments:

- Averted solution or averted cues are not introduced, before being used in section 3.2. What is the difference between averted and preferred? Section 3.2 seems to imply that preferred solutions are computed by optimizing the model using D_diag and averted solutions computed using D_i for a specific property i that is not deemed preferred. Am I correct? If this is the case, it seems unreasonable to be comparing the two solutions directly (eg in Fig 4) since they are computed using datasets of different sizes.
- In Fig 4, it also is not clear what the loss landscape represents. Section 3.2 mentions that it's the loss around averted and preferred solutions, but Figure 4 does not mention which ones are averted and which ones are preferred. Also, Figure 4 says that the two directions of parameter variation were chosen at random. Given the high dimensionality of DNN parameters, it would be much more informative to depict several of these plots per solution, to cover more directions of variation.
- Section 4 provides evidence to the idea that certain cues have higher complexity than others. This is measured by searching for the smallest model that can memorized the training set. Color could be memorized by a model with only 1.2K parameters, and orientations needed 273K parameters. A natural question is then whether large models present less biases towards simpler cues. As the parameter count increases, the solution set for the orientation task must also increase. It would be interesting to see if the model preference for more complex cues would also increase, or if solution sets for simpler cues would still dominate for very large models.
- Although the diagonal dataset construct is appropriate to see which cues are preferred by a model, in practice this construct almost never applies. Real data may present correlations similar to those found in the diagonal construct, but will also contain off-diagonal samples. Thus, an interesting analysis would be to measure how much you'd need to deviate from D_diag to obtain an "averted solution". In other words, if we train a model with (100-p)% diagonal samples and p% off-diagonal samples labeled according to cue i (e.g. labeled for orientation), how likely is the model to correctly predict cue i for increasing values of p?

**Summary Of The Paper:**

The paper conducts a study of which visual cues are preferred by current vision models. The paper designs a training setup with several cues where each cue is equally correlated with the image label. The paper shows that visual cues like color are much easier to be learned by a vision model, than other cues such as orientation and shape. The paper also provides evidence that easy-to-learn cues tend to converge to relatively flat minima and models that prefer these cues are more abundant in parameter space.

**Summary Of The Review:**

The paper is a very interesting read. It provides an interesting and insightful analysis of model preferences for various visual cues. The analysis shows the preference of visual models towards low complexity visual cues, such as color and ethnicity. Given the potential impact of the analysis, I recommend the paper to be accepted. I would nevertheless strongly encourage the authors to address my main comments/concerns.

---

> ### Author Response · Authors · 2021-11-17
> **Response 2/2**
>
> > Section 4 provides evidence to the idea that certain cues have higher complexity than others. This is measured by searching for the smallest model that can memorize the training set. Color could be memorized by a model with only 1.2K parameters, and orientations needed 273 K parameters. A natural question is then whether large models present a less degree of biases towards simpler cues. As the parameter count increases, the solution set for the orientation task must also increase. It would be interesting to see if the model preference for more complex cues would also increase, or if solution sets for simpler cues would still dominate for very large models.
>
> Great question. We believe this is a good chance to clear up our conclusions at different model sizes. Let us consider, for example:
> - Model **S**mall: 10K parameters
> - Model **L**arge: 1M parameters
>
> Model **S** is capable of representing color but not orientation. In the parameter space, there are color-biased solutions but no orientation-biased ones.
>
> Model **L** is fully capable of representing both color and orientation, with both types of solutions in the parameters space. Thus, the reviewer is correct to say that compared to model **S**, the larger model **L** has a greater volume of orientation-biased solutions in the parameter space.
>
> The main setup the paper is concerned with is the models of type **L**, when a model is capable of representing both simple and complex cues. Our empirical and theoretical conclusion is that even with the stated capabilities, such models prefer simple cues. In the parameter space, color-biased solutions are exponentially more abundant than orientation-biased ones (Section 4, $p(f)\lesssim 2^{-a\widetilde{K}(f)+b}$). We expect it to be difficult to reverse models’ natural inclinations towards simplicity by merely increasing the model size.
>
> What is not inferred from our studies is whether the preference to simple cues will become less glaring as the model size increases far beyond **L**. It would be a great finding if there is a continual increase in the relative volume of complex solutions in larger models. But at the same time, one needs to keep in mind that the relative volume of complex solutions is always bound to be exponentially smaller than that of simple solutions (Section 4, $p(f)\lesssim 2^{-a\widetilde{K}(f)+b}$).
>
>
> > Although the diagonal dataset construct is appropriate to see which cues are preferred by a model, in practice this construct rarely applies. Real data may present correlations similar to those found in the diagonal construct, but will also contain off-diagonal samples. Thus, an interesting analysis would be to measure how much you'd need to deviate from D_diag to obtain an "averted solution". In other words, if we train a model with (100-p)% diagonal samples and p% off-diagonal samples labelled according to cue i (e.g. labelled for orientation), how likely is the model to correctly predict cue i for increasing values of p?
>
> Thank you for this interesting comment. The setup of the experiment is indeed purposefully extreme to let us understand the results better. The experiment proposed by the reviewer is very pertinent and practically meaningful. We expect to observe that an averted cue will require more off-diagonal samples (p%) to increase its ranking, and vice versa.
>
> Although pressed for a limited time, we are currently attempting to generate additional results following the reviewer’s interesting thought experiment. We will include them in the supplementary materials should this be possible within the rebuttal deadline.

---

> ### Author Response · Authors · 2021-11-17
> **Response 1/2**
>
> We thank the reviewer for providing a thorough review of the paper, for highlighting relevant and important aspects of this work, and for providing us with valuable feedback to improve its presentation and engage a wider audience.
>
> Below we will address the main comments/concerns in order.
>
> > Averted solution or averted cues are not introduced, before being used in section 3.2. What is the difference between averted and preferred? Section 3.2 seems to imply that preferred solutions are computed by optimizing the model using D_diag and averted solutions computed using D_i for a specific property i that is not deemed preferred. Am I correct? If this is the case, it seems unreasonable to be comparing the two solutions directly (e.g. in Fig 4) since they are computed using datasets of different sizes.
>
> Thank you for asking for clarifications. The notion of preferred versus averted cues (and solutions likewise) is relative. Given a pair of cues A and B, we determine which one is preferred and averted based on a model’s tendency to learn one over the other when trained on the diagonal set $\mathcal{D}_\text{diag}$. We had not previously introduced the terms unequivocally before using them in Section 3.2. We have inserted this definition at the end of Section 3.1 and added appropriate text to clarify the use of “preferred” and “averted” in the manuscript.
>
> It is true that one could often obtain the preferred solution by simply training on the $\mathcal{D}_\text{diag}$. However, as the reviewer has pointed out, this will lead to an imbalance in the number of training data for the preferred and averted solutions, introducing an uncontrolled factor. Instead of doing this, we always use the union of the diagonal and off-diagonal sets, denoted $\mathcal{D}_k$ (Equation 2), for finding both preferred and averted solutions. Though redundant, this ensures a fair comparison. We have updated the paragraph “How to find the averted solutions” in Section 3.2 with this description.
>
>
>
> > In Fig 4, it also is not clear what the loss landscape represents. Section 3.2 mentions that it's the loss around averted and preferred solutions, but Figure 4 does not mention which ones are averted and which ones are preferred. Also, Figure 4 says that the two directions of parameter variation were chosen at random. Given the high dimensionality of DNN parameters, it would be much more informative to depict several of these plots per solution, to cover more directions of variation.
>
> As “averted” and “preferred” solutions are relative concepts, it is not possible to label each cue in Figure 4 as either “averted” or “preferred” without reference points. Instead, we have sorted the cues in the descending order of preference. For example, scale is a preferred cue against orientation but is an averted cue against shape.
>
> It will help the reader to further clarify that we use different datasets, and consequently different loss functions, for finding the solutions versus for plotting the loss landscape in Figure 4. For finding the solutions (the center of the X-Y plane for each plot in Figure 4), we use the union of diagonal and off-diagonal sets, or $\mathcal{D}_k$; see also the response above. For plotting the loss values (loss landscape for each plot in Figure 4), we only use the diagonal set $\mathcal{D}_\text{diag}$, which is shared by all cues. It important to notice that the solutions found with each union set $\mathcal{D}_k$ are still meaningful to the original task $\mathcal{D}_\text{diag}$. The solutions found with $\mathcal{D}_k$ **are** simultaneously the solutions to $\mathcal{D}_\text{diag}$, by Corollary 3. In other words, we have discovered a diverse set of solutions for the task $\mathcal{D}_\text{diag}$ with the help of off-diagonal augmentation $\mathcal{D}_k$ with different $k$’s. Now, since all the plots in Figure 4 represent the loss values for the diagonal set $\mathcal{D}_k$, we can directly compare the loss landscapes across the solutions biased to different cues. Following the reviewer comment, we have revised the Figure’s caption to clarify this important aspect.
>
> The main purpose of Figure 4 was to share our observations and some intuition with the readers. We decided that supplying multiple samples of the 3D plots in a paper was not the best use of the medium. Instead, after sharing the intuition in Figure 4, we have presented a quantitative version of the observations in Section 3.2 and Figure 5, where we measure the breadth of the basin of attraction for each solution type. The breadth is measured in terms of the mean across multiple directions.

---

### Official Review · Reviewer_tkjD · 2021-11-06

**Correctness:** 3
**Technical Novelty And Significance:** 2
**Empirical Novelty And Significance:** 2
**Recommendation:** 6
**Confidence:** 4

**Main Review:**

Strengths:

Overall this is a well-written paper, clearly motivated and carefully constructed. The finding that the number of solutions (i.e. parameter configurations) that rely on preferred cues are also abundant is a good corroboratory result.

The motivations for this study are also exactly what the field needs at this point, given the abundant use of deep convolutional networks for a variety of applications with minimal understanding of its exact decision-making mechanisms.

Though the synthetic task (Wisconsin card sorting) has been widely adopted in the cognitive neuroscience community, their formulation for mainstream ML is nifty and allows for systematic empirical evaluation.

Weaknesses:

Having said that, this paper fails to deliver on its promise of intricate analysis.
(i) There is an inherent question that authors fail to sufficiently address. The selected "cues" for analysis have intrinsically different extraction demands from the stimuli, thus making the argument about them being "equally prevalent" moot. For example, accessing color is more direct (pixel-level information is directly available) than accessing, say, shape (for which a network needs to build sufficiently large receptive fields to understand the global geometry of a scene).

(ii) The experiments on the UTKFace dataset are not convincing. The authors themselves acknowledge that there might be other "shortcut cues" outside of the selected ethnicity and age cues. This brings up a subtle (yet important) question. In naturalistic datasets, how can one determine or interpret a basis set of cues that are truly orthogonal dimensions? The experiments on WCST-ML work because the shape, color, scale, and orientation are by definition orthogonal. For all we know, the "cues" in naturalistic datasets could be abstract and not human-interpretable too.

(iii) Treating the number of model parameters as a proxy for the Kolmogorov complexity of the input-output mapping seems ill-advised. If one trains a parameter-shared recurrent neural network it can perform the same effective computations as a ResNet, but with much fewer parameters (though the task per se has not changed, and by extension the cues). I would appreciate it if the authors can offer some insight into this.

Minor:
(i) Figures need to be made more legible. Axis labels and text insets are barely visible without zooming in.
(ii) Particularly, Figure 4 could be refined further. The caption isn't very descriptive, and the section on "Qualitative view on the loss surface" is a bit misleading. The ethnicity solution doesn't seem to be "characterized with a flatter and wider surface" as compared to age and gender solutions in Fig. 4. If there are more obvious examples, the authors should use them instead.

**Summary Of The Paper:**

The authors propose a framework for studying the tendency of deep neural networks to preferentially adopt "cues". Specifically, they focus on settings where multiple cues are equally likely, though not all of them are equally exploited. To set up such a scenario, they introduce the WCST-ML task, in which the prevalence of cues can be parametrically controlled. They also conduct empirical studies on the more naturalistic UTKFace dataset. The authors introduce a set of metrics, such as path connectivity, attractor basin properties, etc. to analyze cue preferences from a loss landscape perspective. The authors also explain these observations based on the "complexity" of cues.

**Summary Of The Review:**

This is a well-written paper trying to address an important problem. However, as I've expressed in the main review, some of the claims are unsupported and hence my initial assessment. I am willing to update my score if the authors are able to provide a convincing response!

---

> ### Author Response · Authors · 2021-11-17
> **Response 1/2**
>
> We thank the reviewer for the thoughtful and insightful review. We are glad that the reviewer recognises the significance of the topic and the benefit of our approach and analysis. Now, let’s focus on the weaknesses pointed out by the reviewer. The pointed weaknesses are great points too; we have been able to deepen the arguments in the paper thanks to the reviewer’s comments.
>
> > There is an inherent question that authors fail to sufficiently address. The selected "cues" for analysis have intrinsically different extraction demands from the stimuli, thus making the argument about them being "equally prevalent" moot. For example, accessing color is more direct (pixel-level information is directly available) than accessing, say, shape (for which a network needs to build sufficiently large receptive fields to understand the global geometry of a scene).
>
> To start, we’d like to differentiate what we observe from what we control. We have used terminologies like below throughout the paper:
> - equally conducive
> - equally valid
> - equally plausible
> - equally correlates with the targets
> - equally represented
>
> They synonymously indicate the condition that the *degrees of correlation with the target label* for every cue are *identical*. This is what we control. What we observe and analyse, instead, is that there are “different extraction demands” for each cue, as seen by the highly skewed likelihood of each cue being chosen by a deep model.
>
> The reviewer is totally right about the fact that cues do have different intrinsic easiness of accessibility. Our work is focused on exactly this phenomenon. For example, when comparing cues such as color vs shape, color recognition only necessitates reading off pixel values (i.e. “pixel-level information is directly available”), while shape recognition requires something far more complex. So color is indeed easier. However, the question becomes highly non-obvious for other cues, such as shape versus orientation, or in the case of UTKFace, Age vs Gender. Both are high-level concepts. How do we generalize the concept of certain information being “more directly available” to an arbitrary pair of cues?
>
> This is where the mathematical framework comes in. The mathematical tool in our case is the *Kolmogorov complexity (KC)*. We propose a formal way to characterise the “direct availability” of a cue via its readiness to be represented by a simple model. That is, a cue is “directly available” if and only if a small model ($f$ with small $K(f)$) can already effectively memorise the cue pattern (i.e. achieve $\mathcal{L}(f;X,Y)<\delta$); see Equation 6. The benefit of this characterisation is that one can apply the KC concept to an arbitrary cue, as long as we have access to some labelled data for the cue (X-Y pairs).
>
> > In naturalistic datasets, how can one determine or interpret a basis set of cues that are truly orthogonal dimensions? The experiments on WCST-ML work because the shape, color, scale, and orientation are by definition orthogonal. For all we know, the "cues" in naturalistic datasets could be abstract and not human-interpretable too.
>
> Thanks for raising this point. WCST-ML experiments do not require the cues to be orthogonal. The point we try to convey is that deep models prefer certain cues over others. On UTKFace, we have shown that the ethnicity cues are preferred to gender or age, without relying on the orthogonality of the cues. It is also important to point out that, within WCST-ML, human interpretability is also not a requirement. We need a much weaker assumption: that we can put integer cue labels on samples. In the extreme case, we may also define cues A and B on the dataset by uniform-sampling the labels (random noise labels) twice for each input. We may still compare a model’s preference between the cues A and B defined through the respective (X,Y) samples by performing the identical analysis as we have in Section 3. In this work, we chose orthogonal (e.g. shape and color) and interpretable (e.g. gender and ethnicity) cues as an illustrative and meaningful example to the readers, but should labels be available, WCST-ML and its analysis can also be used for an entangled and non-interpretable set of cues. We believe we had not appropriately discussed these points in our previous version, and we have thus updated the paper with this discussion (Section 2.1, below Proposition 1).

---

> ### Author Response · Authors · 2021-11-17
> **Response 2/2**
>
> > Treating the number of model parameters as a proxy for the Kolmogorov complexity of the input-output mapping seems ill-advised. If one trains a parameter-shared recurrent neural network it can perform the same effective computations as a ResNet, but with much fewer parameters (though the task per se has not changed, and by extension the cues). I would appreciate it if the authors can offer some insight into this.
>
> This is a great point. A crucial detail that we failed to emphasise in the submission is that we need to confine the computation of the number of parameters to a single model family. In our case, we have fixed the space of models to *ResNet20 with a varying number of channels* (Section 4, under paragraph “Measuring KCC”). This prevents the ill-defined ordering of network simplicity that arises from comparing e.g. ResNet family versus RNN family, as pointed out by the reviewer. We have emphasised this strategy in the revised manuscript (Section 4, under paragraph “Measuring KCC”).
>
>
> >  Figures need to be made more legible. Axis labels and text insets are barely visible without zooming in. Figure 4 could be refined further. The caption isn't very descriptive.
>
> We have updated the figures as per the reviewer’s comment.
>
> Figure 3, 4, 5, 6, B.1, C.1: Increased font sizes by 20%.
> Figure 4: We have updated the caption with more information.
>
> > The ethnicity solution doesn't seem to be "characterized with a flatter and wider surface" as compared to age and gender solutions in Fig. 4.
>
> One can observe the flatter and wider surface for the ethnicity solution by comparing the size of the blue area against that for the other cues. This is more visible when viewed from the top (right plots for each cue). Albeit minute, we do see that the blue area is slightly wider for ethnicity than age and gender. We agree our description is quite strong, especially given the high subjectivity of the observation. We have fixed the description in the manuscript (Section 3.2, under paragraph “Qualitative view on the loss surface”). As a means to complement the subjectivity of qualitative observations, we have introduced *quantitative metrics* measuring the width of the basin of attraction in Section 3.2 and Figure 5.

---

### Author Response · Authors · 2021-11-17
**Summary of changes in the revision.**

We summarise the changes made in the revision. In the updated PDF, the changes are marked with **red text**.

### Section 1 (Introduction)

* **Figure 1**. Revised the caption and the related paragraph to clarify the conclusion from the illustration of WCST-ML. [Reviewer QJQp]


### Section 2 (Setup)

* **Section 2.1**, below Proposition 1: Elaborate on the conditions for the cues to be analysed with WCST-ML. [Reviewer QJQp]


### Section 3 (Observations)

* **Section 3.1, under paragraph “Models adopt cues with uneven likelihood”**: Fixed the description for Figure 3 to correctly capture the observation. [Reviewer QJQp]

* **End of Section 3.1**: Inserted definitions for “preferred” and “averted” cues. [Reviewer s3vA]

* **Section 3.2, under paragraph “How to find the averted solutions”**: Explain how we match the amount of training data for preferred and averted solutions. [Reviewer s3vA]

* **Section 3.2, under paragraph “Qualitative view on the loss surface”**: Downtoned the description. [Reviewer tkjD]

* **Figure 4**: Updated the caption with more information. [Reviewer tkjD] Clarified the underlying dataset $\mathcal{D}_k$ based on which we plot the loss values. [Reviewer s3vA]

* **Figure 6**: Included description of markers and curves in the caption. [Reviewer QJQp]


### Section 5 (Discussion and Conclusion)

* **Paragraph “Toolbox contribution”**: Inserted a discussion on our toolbox contributions. [Reviewer QJQp]


### Overall

* **Figure 3, 4, 5, 6, B.1, C.1**: Increased font sizes by 20%. [Reviewer tkjD]

---

### Decision · Program_Chairs · 2022-01-20

**Decision:**

Accept (Poster)

**Comment:**

The reviewers were split, with one of them leaning towards rejection, primarily due the (perceived) limited impact of the study. I tend to agree with the other reviewers that this paper provides an interesting and original framework for analysis of learning models, and while there are substantial shortcomings, they are outweighed by the positives (including the promise this approach may hold for analysis of learning in more realistic scenarios). I therefore recommend acceptance, if space in the proceedings allows.